# The Link between Diabetes, Pancreatic Tumors, and miRNAs—New Players for Diagnosis and Therapy?

**DOI:** 10.3390/ijms241210252

**Published:** 2023-06-16

**Authors:** Małgorzata Kozłowska, Agnieszka Śliwińska

**Affiliations:** 1Student Scientific Society of Civilization Diseases, Medical University of Lodz, 251 Pomorska, 92-213 Lodz, Poland; kozlowska.malgosia99@gmail.com; 2Department of Nucleic Acid Biochemistry, Medical University of Lodz, Pomorska 251, 92-213 Lodz, Poland

**Keywords:** pancreatic cancer, diabetes mellitus, microRNAs, biomarkers

## Abstract

Despite significant progress in medicine, pancreatic cancer is one of the most tardily diagnosed cancer and is consequently associated with a poor prognosis and a low survival rate. The asymptomatic clinical picture and the lack of relevant diagnostic markers for the early stages of pancreatic cancer are believed to be the major constraints behind an accurate diagnosis of this disease. Furthermore, underlying mechanisms of pancreatic cancer development are still poorly recognized. It is well accepted that diabetes increases the risk of pancreatic cancer development, however the precise mechanisms are weakly investigated. Recent studies are focused on microRNAs as a causative factor of pancreatic cancer. This review aims to provide an overview of the current knowledge of pancreatic cancer and diabetes-associated microRNAs, and their potential in diagnosis and therapy. miR-96, miR-124, miR-21, and miR-10a were identified as promising biomarkers for early pancreatic cancer prediction. miR-26a, miR-101, and miR-200b carry therapeutic potential, as they not only regulate significant biological pathways, including the TGF-β and PI3K/AKT, but their re-expression contributes to the improvement of the prognosis by reducing invasiveness or chemoresistance. In diabetes, there are also changes in the expression of microRNAs, such as in miR-145, miR-29c, and miR-143. These microRNAs are involved, among others, in insulin signaling, including IRS-1 and AKT (miR-145), glucose homeostasis (hsa-miR-21), and glucose reuptake and gluconeogenesis (miR-29c). Although, changes in the expression of the same microRNAs are observed in both pancreatic cancer and diabetes, they exert different molecular effects. For example, miR-181a is upregulated in both pancreatic cancer and diabetes mellitus, but in diabetes it contributes to insulin resistance, whereas in pancreatic cancer it promotes tumor cell migration, respectively. To conclude, dysregulated microRNAs in diabetes affect crucial cellular processes that are involved in pancreatic cancer development and progression.

## 1. Introduction

One of the most common gastrointestinal cancers apart from stomach or colorectal cancer is pancreatic cancer. Considering the localization in the pancreas, it is divided into two types, the most commonly occurring exocrine pancreatic cancer, comprising adenocarcinoma, and neuroendocrine pancreatic cancer, respectively [1]. Despite significant progress in medicine, pancreatic cancer is detected at its advanced stages resulting in a poor prognosis for patients. The largest percentage of diagnosed patients refers to advanced stages with distant metastases, and for these patients a five-year survival rate is less than 5% and a median time from diagnosis is equivalent to around six months [2,3]. Pancreatic cancer risk factors include age, family history, and gene mutations, such as in the *BRCA2* gene, or hereditary syndromes including hereditary pancreatitis [4]. Another risk factor is lifestyle, which is a common factor in the development of both diabetes and pancreatic cancer, with diabetes itself also carrying the risk of developing cancer. An unhealthy lifestyle includes alcohol drinking, smoking, and diet, thereby leading to the development of obesity. Both acute and chronic pancreatitis are crucial risk factors of pancreatic cancer development. Pancreatitis as an inflammatory disease is a source of chronic inflammation leading to pancreatic calcifications and exocrine insufficiency [5]. Acute pancreatitis may constitute an early symptom of pancreatic cancer [6]. Chronic pancreatitis is accompanied with the downregulation of the tumor-suppressing genes, such as *p16*, *TP53*, and *SMAD4* (mothers against decapentaplegic homolog), respectively, along with the upregulation of the oncogenic *KRAS* (Kirsten rat sarcoma virus), *TNF-α* (tumor necrosis factor alpha), and *NF-kB* (nuclear factor kappa B) genes [5].

microRNAs (miRNAs) are small, non-coding molecules involved in RNA silencing and the post-transcriptional regulation of gene expression. They comprise 1–2% of all genes in worms, flies, and mammals [7]. They participate in many biological processes including cell cycle and differentiation regulation, for example the development of muscle and nervous system, they inhibit important cycle regulators, including *CDK1* and *CDK2* (cyclin-dependent kinase) [8]. microRNAs are also engaged in the regulation of the immune response, angiogenesis, and apoptosis [8]. Their dysregulation is one of the factors contributing to many diseases, including pancreatic cancer. The role of microRNAs in the diagnosis and therapy of pancreatic cancer is not yet well understood, so numerous studies are being conducted in this area. Therefore, the aim of this review is to present the latest findings concerning the participation of microRNAs in the pathogenesis, diagnosis, and therapy of pancreatic cancer.

### 1.1. Epidemiology of Pancreatic Cancer 

Data obtained from the WHO and GLOBOCAN 2020 shows that among all cancers, pancreatic cancer is 12th in terms of the incidence and seventh in terms of mortality for both sexes and all ages, respectively. The incidence rates of pancreatic cancer are similar among women and men, with a slight advantage among men—5.5 per 100,000 cases in men, and 4 per 100,000 cases in women, respectively [9,10]. According to the International Agency for Research on Cancer, an agency of the World Health Organization, more than 495,000 people developed pancreatic cancer in 2020, with more than half of these cases occurring in Asia and Europe, as shown in Figure 1. There have been several reasons attributed to this finding, but they are not entirely clear. In addition to factors such as obesity or smoking, the higher incidence of pancreatic cancer observed in countries from Asia and Europe has also been influenced by the extending lifetime that is measured by the Human Development Index. In countries with a longer life expectancy, there are older populations, and thereby pancreatic cancer is more common as it occurs more frequently in older people. Higher incidences in countries of Asia and Europe also result from the availability and quality of diagnostic measures that is different among the particular continents [11]. Statistics conducted by Cancer Research UK have shown that age-specific incidence rates rise from the ages of 35–40, but also that the peak incidence of pancreatic cancer occurs in the 7th and 8th decades of life. There is a general trend towards higher incidence rates of pancreatic cancer observed in developed countries compared to developing countries, which suggests that the large differences in this cancer incidence between these countries may be due to environmental factors [12]. Several syndromes contribute to an increased risk of pancreatic cancer in the high-risk group, for example, in the case of family history, the risk of developing pancreatic cancer increases with the increasing number of first-degree relatives diagnosed with PC (pancreatic cancer) [13]. Further examples of comorbidities include hereditary pancreatitis, cystic fibrosis, and hereditary breast ovarian cancer syndrome [13]. 

### 1.2. Pathophysiology

The pathophysiology of pancreatic cancer is complex and multi-factorial in nature. Risk factors are classified as modifiable and non-modifiable, as presented in Table 1. 

Smoking is considered as the strongest causative factor of numerous cancers, including pancreatic cancer. The pancreas is indirectly exposed to the harmful effects of tobacco smoke containing multiple carcinogens that reach the organ primarily through the blood [14]. It has been recognized that tobacco smoke contains more than 4000 different chemical compounds with proven roles in the process of carcinogenesis, such as nitrosamines, including NNK (nitrosamine ketone) [15]. The product formed after NNK penetration, NNAL (4-(methylnitrosamino)-1-(3-pyridyl)-1-butanol), has been shown to induce pancreatic cancer in hamster and rat animal models [16]. NNK is metabolized by cytochrome P450 enzymes, cyclooxygenases and lipoxygenases and created metabolites after binding to DNA form adducts that are able to form point mutations in numerous genes, including suppressors and protooncogenes e.g., in the *KRAS* gene, which is considered as one of the most commonly mutated in pancreatic cancer [16]. 

Another modifiable risk factor for pancreatic cancer is alcohol consumption. Both ethanol and acetaldehyde that is formed during metabolism of the alcohol, are considered to be compounds with strong carcinogenic effects. In addition, alcohol is also metabolized in the liver by cytochrome P450, which results in the formation of reactive oxygen species (ROS). The latter ones are responsible for oxidative stress induction and the related oxidative damage to macromolecules, such as DNA, proteins, and lipids. Oxidative cellular damage can be removed through various pathways, but they also activate the NF-kB pathway that increases the production of proinflammatory cytokines and the formation of inflammation [17]. Ethanol is also metabolized by the non-oxidative pathway producing FAEEs (fatty acid ethyl esters) which damage pancreatic stromal cells and increase the sensitivity of lysosomes to release endogenous hydrolases which convert trypsinogen to trypsin, causing pancreatic damage and ultimately acute pancreatitis [17,18]. 

An unhealthy diet leading to obesity is also a risk factor for pancreatic cancer. Unhealthy diets include foods high in sodium and sugar, processed food, or high-fat meals at the same time, with a small amount of fruit and vegetables, dairy, or low-fat products. Moreover, the lack of physical activity also leads to obesity. Physical activity increases the sensitivity to insulin, during muscle contraction, the skeletal muscle enhances glucose uptake into the cells and reduces intra-abdominal fat, which is known as a risk factor for both insulin resistance and obesity [19]. Obesity leads to the metabolic dysfunction of adipose tissue that as an endocrine organ releases different proinflammatory cytokines (IL6, TNF, Il-1), adipokines (leptin, adiponectin), or free fatty acids, and thereby creates conditions that facilitates cancer development and progression [20]. The mechanisms underlying the metabolic disorders of adipose tissue is complicated and lead to changes in both endocrine and paracrine signaling. The altered body fat physiology is responsible for chronic inflammation that plays an important role in the proliferation, invasion, and metastasis of cancer cells [20]. In addition, obesity is also a risk factor for the development of type 2 diabetes that further accelerates the formation of conditions that are favorable for pancreatic cancer development, such as hyperglycemia, insulin resistance, oxidative stress, and low-grade chronic inflammation [21]. Prospective studies have reported that peripheral insulin resistance confers an increased risk for pancreatic cancer [22]. Prediabetes, a condition in which glucose metabolism is impaired, but glycated hemoglobin levels are not high enough to diagnose diabetes, can also promote the development of diabetes and then cancer, including pancreatic cancer [23]. 

Diabetes, especially type 2 that results from a sedentary lifestyle, another pancreatic cancer risk factor, is a scourge of the modern world. According to the World Health Organization’s World Diabetes Report, the number of people with diabetes increased from 108 million in 1980 to 422 million in 2014, respectively [24]. Diabetes mellitus is a chronic disease associated with elevated blood glucose levels caused by insulin resistance, or damage to the β-cells of the pancreatic islands and impaired secretion of the hormones. There are many types of diabetes (Table 2), and recently pancreatic type 3c diabetes has been of great interest to researchers. It has been classified as secondary diabetes, referring to the impairment of pancreatic endocrine function associated with exocrine damage to the pancreas as a result of acute, recurrent, and chronic pancreatitis [25]. Chronic hyperglycemia, defined as a state of elevated blood glucose levels, contributes to oxidative stress, which in turn causes an increased release of superoxide anion radicals from the mitochondrial respiratory chain. The anion radicals cause the formation of further reactive ROS, such as H_2_O_2_ (hydrogen peroxide) and ONOO- (peroxynitrite). Another mechanism that generates oxidative stress is the activation of the polyol pathway, the hexosamine pathway, and the process of the formation of glycation-end products [26]. In the polyol pathway of glucose metabolism, glucose is reduced to sorbitol, which is a hydrophilic alcohol, which does not have the ability to diffuse through lipid membranes, thereby causing the hypertonicity of cells, increase in osmotic pressure, and capillary damage [27]. Additionally, there is an increased amount of NADPH (nicotinamide adenine dinucleotide phosphate), a redox imbalance, and an increased ROS production in the cells. The hexosamine pathway, accompanied with hyperglycemia, contribute to the overproduction of mitochondrial superoxide, inhibition of GAPDH (glyceraldehyde-3-phosphate dehydrogenase), and the influx of phosphorylated glucose, which leads to ROS stimulation in the mitochondria and disrupts mitochondrial respiration as a result [27]. During non-enzymatic glycation, reducing sugars co-create unstable Schiff bases, which form irreversible and stable glycation end-products [27]. Amidst this process, a large amount of free radicals are generated and released. As a consequence, this state leads to the damage of both the structures and functions of proteins, and further stiffening of the blood vessels [27]. The resulting oxidative damage and diminished efficacy of the antioxidant defense system creates conditions that facilitate vascular damage and the initiation of tumorigenesis.

One of the non-modifiable risk factors is age. Pancreatic cancer mostly affects the elderly, with the peak incidence falling in the seventh and eighth decades of life [14]. According to data presented by the National Cancer Institute, as many as 21.5% of patients aged 55–64 suffer from pancreatic cancer. In younger patients (45–54 years of age), this type of cancer occurs with a frequency of 7.4%, and in the age group 35–55 it is 1.9%, respectively. 

Pancreatic cancer is also an inherited disease. It was estimated that familial pancreatic cancer accounts for approximately 3–8.4% of all cases of this disease with at least two first-degree relatives [28,29]. The mutations in the following genes were found to be crucially connected with an increased incidence and progression of pancreatic cancer, protooncogenes such as *KRAS*, *erbB-2* (receptor tyrosine kinase 2), and *c-myc* (cellular myelocytomatosis oncogene) and suppressor genes including *CDKN2A* (cyclin dependent kinase inhibitor 2A), *TP53*, and *SMAD4*, respectively [30,31]. It should be highlighted that diseases such as cystic fibrosis, Peutz–Jeghers syndrome, and Lynch syndrome markedly increase the risk of pancreatic cancer [3]. One of the risk factors of pancreatic cancer is cystic lesions of the organ, such as intraductal papillary mucinous neoplasms (IPMN). IPMNs are defined as intraductal epithelial neoplasms of mucin-producing cells. They are usually resectable, but they have the potential for malignant transformation [32]. It has been shown that in both IPMN and PDAC, there is an increased level of miR-155, and a decreased level of miR-101, respectively [33,34]. Caponi et al. showed that significantly overexpressed miR-155 was in the neoplastic epithelium of IPMN, and in addition, it had a markedly higher level in invasive lesions compared to non-invasive lesions, suggesting that this microRNA can be used as an early marker of malignant transformation [35]. miR-155 represses tumor protein 53-induced nuclear protein-1, which is a proapoptotic stress-induced p53 target [36]. Furthermore, there was a significantly reduced level of this protein found in PDAC, thereby revealing another link between miR-155 and IPMN and PDAC [37]. In turn, miR-101 is downregulated in invasive IPMN compared to non-invasive IPMN and healthy tissue, thereby indicating its involvement in tumor invasion [35]. One of the targets of miR-101 is the *EZH2*, which controls DNA methylation. The loss of miR-101 has been suggested as a trigger for the adenocarcinoma sequence of IPMN by the upregulation of *EZH2* [37].

### 1.3. Diagnosis and Therapy

Despite significant developments in diagnostic methods, pancreatic cancers are diagnosed mainly at the advanced stages of the disease due to late-appearing and non-specific symptoms. The most commonly reported symptoms by patients are lower abdominal pain, weight loss, nausea, and “painless jaundice” [33]. The symptoms are so non-specific that they are often confused with other diseases, such as cholecystitis, cholelithiasis, or duodenal and gastric ulcers [10]. 

The main methods of pancreatic cancer diagnosis are imaging methods, including computed tomography (CT), magnetic resonance imaging (MRI), positron emission tomography (PET) and endoscopic ultrasound (EUS) [38]. These methods are sensitive enough to detect pancreatic tumors and to assess local or distant metastases. They are also required for the collection of biological materials from the patient using fine-needle biopsy for further histological evaluations of the affected tissue [38]. However, imaging methods are not sufficient in detecting tumors with a size of <1 cm, not effective in detecting small metastases or changes in the peritoneum, and their price for utilization is high [39,40]. Therefore, in order to aid diagnosis, especially in the early stages of the disease, it is necessary to develop diagnostic methods based on biomarkers specific to this type of tumor. Currently, a few diagnostics markers are available, namely circulating cancer cells in the blood CTCs (circulating tumor cells), epigenetic markers such as ADAMTS1 (disintegrin A and metalloproteinases with thrombospondin motifs), tumor markers such as CA19-9 (cancer antigen 19-9), autoantibodies such as anti-MUC1 (mucin 1), ENOA1/2 (alfa-enolase), or against RAD1 (RAD1 checkpoint DNA exonuclease) for the early diagnosis of the disease and assessment of prognosis, and microRNAs such as miR-196a, miR-221, and miR-155 [39,41]. The most commonly used method is the detection of CTCs, and it is characterized with a high sensitivity and specificity [42]. Another method accepted by the FDA (Food and Drug Administration) is the detection of CA19-9 antibodies, but this has a low sensitivity (70%) and specificity (87%) [43]. Therefore, considering the insufficient number of markers used to detect pancreatic cancer, it is necessary to continue searching for new diagnostic methods. The greatest advantage of using molecular diagnostic markers is their non-invasiveness, high specificity, and opportunity for the diagnosis of pancreatic cancer at its earlier stages. 

Due to the late detection of the disease, in most cases the therapy is associated with a resection of the pancreas, followed by radiotherapy or adjuvant chemotherapy [44]. However, the frequent complications or unresectability of the tumor necessitate the search for new therapies in the treatment of this cancer. In addition to the use of a new combination of different chemotherapeutic agents, the IRE (irreversible electroporation) technique raises great hope. With this method, short pulses of high-frequency current destabilize the cell membrane and allow the penetration of proteins or drug molecules into the cell [3,45]. This method is safe as it does not damage the pancreatic ducts or vessels, and additionally enhances the effects of chemotherapy and feasible primary local treatments in unresectable pancreatic cancer [46]. Another treatment approach is immunotherapy based on modified T cells (CART-T) or the modulation of the response by myeloid cells and their use for therapeutic purposes [47]. Myeloid cells are required for sustained MAPK (mitogenic-activated protein kinase) signaling in pancreatic epithelial cells during the onset of carcinogenesis [48]. These cells regulate the expression of the immune checkpoint PD-L1 (programmed cell death ligand) and then suppress anti-tumor immune responses mediated by CD8+ cells [48]. Another myeloid cell-based approach involves inhibiting certain cytokines, chemokines, or their receptors, including CXCR2 (C-X-C motif chemokine receptor 2) and targeting checkpoints, thereby increasing T-cell filtration into the tumor and effectively killing tumor cells as a result [47]. This method is not common due to the high phenotypic and functional heterogeneity of this cell population. Regrettably, presented methods, similar to IRE, are quite expensive and their effectiveness and safety are being assessed in clinical trials. CART-T therapy, although expensive, is gaining popularity among the specialists. 

To sum up, despite the fact that new methods for the diagnosis and treatment of pancreatic cancer have been introduced, the low survival rate among patients is still observed, therefore leading to the necessity to develop new methods. In line with the present trends, scientists are focusing on personalized medicine, including the use of organoids, the microenvironment of cancer, stem cells, and microRNAs [49,50]. 

## 2. microRNA—An Overview

MicroRNAs are non-coding, single-stranded small RNAs, with an average length of 19–25 nucleotides. They are involved in many biological processes, such as cell growth, proliferation, differentiation, and organogenesis, and their main mechanism of action is to regulate gene function either by the degradation of mRNA or through the inhibition of mRNA translation [51]. 

The biogenesis of microRNA consists of a series of processes occurring in both the nucleus and in the cytoplasm, and genes for microRNAs are found in both exons, introns, and untranslated sequences, thereby allowing for the simultaneous formation of microRNA transcripts and mRNAs [52]. One of the first stages of biogenesis is the post-transcriptional formation of primary transcripts (pri-miRNA) with a cap at the 5’ end and a poly(A) tail, which are then capped and spliced by the nuclear protein DGCR8 (DiGeorge syndrome critical region) combined with the Drosha enzyme which contains two RNase III domains, each of which cleaves one strand of the dsRNA [53]. The pri-miRNA formed has a characteristic hairpin-shaped structure and transport proteins such as Exportin 5 and is then transported from the nucleus to the cytoplasm, where the next step in microRNA formation takes place. The processing of the pre-miRNA is performed by the enzyme Dicer and involves the cutting and removing of the terminal loop, resulting in the formation of a mature microRNA–microRNA duplex, in which one strand is the guide strand and the other is the passenger strand, respectively [52,53]. The Argonaute family proteins are the core components of the RNA-induced silencing complex (RISC), which attaches to the mature microRNA, and this activated complex is able to cause the repression of the translation and the degradation of the target mRNA [52,54]. Both the leading and passenger strands are able to participate in the regulation of expression such that one of the microRNA strands is complementary to the target mRNA, and this involves the negative regulation of gene expression at the post-transcriptional level [53]. The complementarity between the microRNA and mRNA does not have to be complete, meaning one microRNA could regulate many genes as its targets, and reversely one gene may be targeted by many microRNAs [55]. 

microRNAs are involved in the regulation of many physiological processes required for the function of the organism, including differentiation and apoptosis. However, they are also involved in the processes associated with the development of several diseases, such as angiogenesis or carcinogenesis. In the case of carcinogenesis, the changes in the expression of microRNAs during this process have been observed, and the diversity of the genes regulated by microRNAs makes it possible for them to behave both as oncogenes and as transformation suppressors [50]. The alterations in the expression of various microRNAs have also been observed in pancreatic cancers, as depicted in Figure 2. They have been associated with pancreatic cancer progression and metastasis, suggesting that microRNA expression profiling may serve as both diagnostic and prognostic biomarkers [56].

Studies have shown that Dicer, involved in the biogenesis and activity of microRNA, participates in pancreatic cancer development. Dicer’s function is important throughout the development of the pancreas, as it affects the endocrine functions of this organ [57,58]. Studies conducted on mice in which knockout Dicer was used revealed that these animals had a radical reduction in the ventral pancreas, and a decrease in the overall epithelial contribution to the dorsal pancreas [58]. Staining for specific hormones showed a significant loss of β-cells responsible for insulin production [58]. In cancers, including pancreatic cancer, there is an altered expression of various microRNAs, and this may be due to the altered expression of the enzyme Dicer by targeting ERK/Sp1 signaling, which can directly contribute to the initiation, growth, and progression of the tumor [59]. Research conducted on mice have shown that the reduced or even complete knockout of expression of Dicer accelerates KRAS-driven acinar dedifferentiation and leads to the loss in the polarity of cluster cells, which initiates both epithelial-to-mesenchymal transitions (EMT) and cluster metaplasia to ductal, a process that has been shown to precede and promote the formation of pancreatic cancer precursors [60]. Other studies have shown a positive correlation between the high expression of the enzyme Dicer and resistance to gemcitabine, a cytostatic drug used in cancer therapy [59,61]. Interestingly, study conducted on platelets collected from patients with diabetes revealed that Dicer levels were significantly reduced compared to the healthy patients (79 ± 3%), what may suggest an association between diabetes and pancreatic cancer, but there is still too little data on this issue [62].

One of the activities that microRNAs exhibit in tumorigenesis is their participation in initiation stage through dysregulation of apoptosis, also termed as programmed cell death. Under physiological conditions, in response to DNA damage, either DNA repair or apoptosis is induced, and thus cellular homeostasis is maintained. It was demonstrated that microRNAs participate in the dysregulation of apoptosis in cancer. One of them is miR-155, that overexpression in pancreatic cancer resulted in the down-regulation of p53 protein, thereby inhibiting apoptosis [36].

The next stage of carcinogenesis is tumor progression stage connected with dysregulation of the cell cycle. The later one is regulated by numerous checkpoints, oncogenes, and tumor suppressors. In cancer, excessive cell proliferation is observed, which is associated, among others, with the overexpression of oncogenic microRNAs, such as miR-21. Its upregulation negatively affects the expression of tumor suppressors, such as *PTEN* (Phosphatase and Tensin Homolog), *SMAD7* (mothers against decapentaplegic homolog), *PDCD4* (programmed cell death 4), or *KRAS*, which are involved in the essential pathways connected with the regulation of the proliferation, growth, and transformation of epithelial cells [63]. Another microRNA related with increased cell proliferation is miR-424-5p, whose up-expression was observed in pancreatic cancer, evoked the decreases in SOC6 (suppressor of cytokine signaling) protein levels and increased ERK (extracellular signal-regulated kinase) pathway activity [64]. miR-27a, an oncogenic microRNA, is also significantly upregulated in pancreatic cancer, and its inhibition has been shown to reduce tumor cell growth and proliferation by inducing the tumor suppressor Spry2 (Sprouty) [65]. It has been observed that in pancreatic cancers there is also the downregulation of several suppressor microRNAs, including miR-124, miR-203, and miR-96. miR-124 regulates the expression of the *RAC1* (Rac family small GTPase 1) oncogene and inhibits tumor cell proliferation. However, it has been shown that in pancreatic cancer reduced expression of miR-124 interferes with this process, and it was associated with a poor survival among patients with pancreatic cancer [63]. In turn, the decreased expression of miR-203 leads to the transition to the G1 phase of the cell cycle, and consequently increases the proliferation of the cancer cells [66]. 

Another important role that can be attributed to microRNAs is their involvement in the regulation of invasion and the formation of metastases. One of the elements of cancer cell biology is their invasiveness, which is the ability to migrate from the original tumor mass and then enter the bloodstream or lymphatic system to settle in a new, distant place and proliferate from there in an uncontrolled way, thereby creating metastases [67]. Such metastases significantly reduce the prognosis for patients and contribute to an increased mortality. The key step in invasion and metastasis is the epithelial-to-mesenchymal transition, which enables cancer cells to migrate. microRNAs are also engaged in the EMT transition. Upregulation of miR-10a was found to promote pancreatic cancer metastasis, and this was found to be caused by the decrease in the expression of the HOXB1 (homeobox protein) and HOXB3 genes, which encode a highly conserved family of transcription factors necessary for the proper course of morphogenesis [68]. Another microRNA associated with metastasis is miR-34b, which negatively regulates the expression of Smad3 protein, the metastasis promoter. Reduced levels of this microRNA was observed in pancreatic cancers [64]. It has also been shown that members of the miR-200 family and the ZEB1 (zinc finger E-box binding homeobox) and SIP1 (Smad-interacting protein-1) proteins regulate each other to form a feedback loop, thereby controlling the epithelial-to-mesenchymal transition. The decrease in the expression of the miR-200 family was found to induce the EMT [66]. 

### The Role of microRNAs in the Therapeutic Approaches 

The utility of microRNAs in therapeutic approaches are based on several of their features. Firstly, microRNAs are smaller than proteins, meaning they can be easily introduced into the cell and also easily isolate from the biological samples, such as serum, saliva, or milk [69]. Secondly, microRNAs are secreted by different cell types and are transported to the target site by extracellular vesicles, meaning they are able to regulate the tumor microenvironment as a result. Thus, knowing the biology and the mechanisms of action of microRNAs, the therapeutic approach focuses primarily on either inhibiting the oncogenic microRNAs, or restoring the expression of suppressor microRNAs. Another therapeutic approach is the use of anti-miR oligonucleotides, which sequester the mature microRNA leading to the direct functional inhibition of the microRNA and the depression of their targets [70,71]. These features have made these non-coding RNAs as an interesting target for many scientists and clinicians.

The use of microRNAs in therapy was demonstrated by Guo et al. [72]. miR-15 acts as a tumor suppressor and alters cell cycle control. *In vivo* studies have shown that miR-15 in combination with 5-FU improves survival, either alone or in combination with gemcitabine [72]. In in vitro studies conducted on three pancreatic cancer cell lines (AsPC-1, PANC-1, and Hs 766T, respectively), the 5-FU-miR-15 complex was found to inhibit the proliferation of these pancreatic cancer cells, and additionally sensitize the cells to gemcitabine, which thereby enhanced the therapeutic effects [72]. The effects of miR-15a are mediated through the regulation of several important target genes, including checkpoint kinase 1 (*Chk1*) and *Yap-1* [72]. Both genes are elevated in pancreatic ductal adenocarcinoma and are good target candidates for therapeutic development in PDAC (pancreatic ductal adenocarcinoma) [72]. The administration of miR-15 with 5-FU resulted in the inclusion of the 5-FU-miR-15 complex in the RISC complex, and the suppression of Yap-1 and Chk1 resulting in cell cycle arrest, apoptosis, and the attenuation of EMT [72]. To assess how the 5-FU-miR-15 complex affected metastasis and chemoresistance, researchers have created mouse models of pancreatic cancer metastasis and then implemented therapy with 5-FU-miR-15. The results showed that this complex significantly inhibited the growth of a metastatic tumor at a dose fifteen times lower than gemcitabine, and the inhibitory effect was further enhanced in combination with gemcitabine [72]. In addition, Guo et al. found that resistance developed in a group of mice treated with gemcitabine alone, but not in mice treated with the 5-FU-miR-15 complex, which was connected with a significantly longer survival [72]. 

microRNAs can also be used to monitor the course of therapy. An example of such a microRNA is miR-107, which under physiological conditions is a tumor suppressor. However, its reduced levels were identified in the plasma of people with pancreatic cancer. A study by Imamura et al. found that following surgery to remove the pancreatic tumor, miR-107 level increased significantly, which was associated with health improvement [73]. Meijer et al. demonstrated the effectiveness of miR-181a-5p in monitoring FOLFIRINOX therapy in patients with pancreatic cancer. After the treatment, there was a significant decrease in miR-181a-5p in the plasma, which was associated with a better prognosis for patients [74]. 

Currently, the use of microRNAs in treatments for pancreatic cancer are not approved for use, and clinical trials are underway to see whether microRNAs can be utilized in the detection and therapy of both pancreatic cancer. These attempts include the creation of a biobank of pancreatic cancer tissue and plasma from patients suffering from pancreatic cancer, pancreatitis, and normal pancreas for early detection. However, despite positive data from the experimental treatment of pancreatic cancer with microRNAs, there are still too few studies to include microRNA in the therapy as an agent or as a marker for therapy monitoring.

## 3. Diabetes and Pancreatic Cancer

Diabetes is one of the civilizational diseases characterized by a constantly growing number of incidences. According to the World Diabetes Federation (IDF) data from 2021, 537 million adults live with diabetes (often undiagnosed), which is 1 person per 10 individuals, and by 2030 this number will be expected to increase to 643 million [75]. Diabetes and its late diagnosis are not only burdensome for the patient but is also a burden financially for health care systems, as it is associated with a significant increase in the costs of diabetes and its complicative treatment. According to the generally accepted definition, diabetes mellitus is defined as a group of heterogeneous metabolic disorders which are caused by an impaired or absent insulin secretion, with a common feature being elevated blood glucose levels [76]. In most people with diabetes, clinical diagnosis occurs following a long delay, which is associated with the fact that the initial period of the disease is asymptomatic, and progressive hyperglycemia causes not only microvascular changes, but also contributes to both morphological and functional changes in tissues, leading to the impairment of the key organs [77]. Diabetes mellitus is divided into several types based on the mechanism of development, age of onset, and method of treatment, as shown in Table 2.

Diabetes mellitus is diagnosed when fasting plasma glucose (FPG) is ≥126 mg/dl or following the oral glucose tolerance test (OGTT) showing blood glucose levels of ≥200 mg/dl, respectively [78]. Another diagnostic tool is the measurement of HbA1c (≥48 mmol/mol), i.e., glycated hemoglobin, which is also used to control the metabolic compliance in diabetic patients. It reflects an average glycemia over a period of 3 months [79]. 

In diabetes, there is no universal schedule of treatment due to the different types and forms of this disease. In type 1 diabetes, the most common form of treatment to control glycemia is the use of insulin in the form of injections or special insulin pumps. In type 2 diabetes, treatment begins with changes in nutritional habits and physical activity, as this type of diabetes is associated with lifestyle-related obesity. Type 2 diabetes patients frequently have insulin resistance, and if these lifestyle changes do not improve glycemia, patients are then treated with metformin, which is the most commonly used oral hypoglycemic drug. Metformin is the first-line drug, which inhibits glucose production in the liver and increases insulin action in the muscles [80]. Other drugs are also used, such as α-glucosidase inhibitors, which prolong the time of the digestion of carbohydrates, incretins that reduce glucagon secretion, or SGLT-2 (sodium-glucose cotransporter), which contributes to the elimination of glucose with the urine and thereby lower blood glucose levels [80]. In other types of diabetes, therapy is matched to the patients, and depends, among other things, on comorbidities or the patient’s state of health.

### 3.1. Relationship between Diabetes and Pancreatic Cancer

Diabetes can be both a risk factor and an early sign of pancreatic cancer [81]. Both diseases share common risk factors, such as obesity and genetic factors, as depicted in Figure 3. Studies have shown that long-term type 2 diabetes is associated with a 1.5–2 times higher risk of developing pancreatic cancer [82]. One of the mechanisms affecting the induction of the tumorigenesis process in diabetes is the increase in the bioavailability of the insulin-like growth factor-1 (IGF-1), which under standard conditions is responsible for lowering blood glucose levels and simultaneously diminishing insulin levels [83]. The insulin resistance present in type 2 diabetes along with the accompanying high insulin levels increase the synthesis of IGF-1. IGF-1 is bounded by the insulin-like growth factor-binding protein 1 (IGFBP), thus reducing the level of the latter. As IGFBP participates in the regulation of the cell cycle, its reduced level in diabetes leads to the inhibition of apoptosis, stimulation of cell proliferation, and the enhancement of angiogenesis [84,85]. Hyperglycemia is a non-physiological condition that causes the activation of several signaling pathways, including the activation of a metabolic pathway involving the DAG (diacyloglycerol)-PKC (protein kinase C) -NADPH oxidase (nicotinamide adenine dinucleotide phosphate) [86]. PKC has been associated with vascular alterations, such as increases in permeability, extracellular matrix synthesis, angiogenesis, and cytokine activation and inhibition [87]. DAGs are lipid signaling messengers which impair homeostasis caused by hyperglycemia and contributes to β-cell dysfunction as a consequence [88]. High concentrations of glucose and non-esterified fatty acids cause NADPH overproduction, and this together with high levels of PKC and DAG cause the production of ROS (reactive oxygen species) and oxidative stress [89,90]. Increased productions of ROS and the accompanying oxidative stress induces glucotoxicity leading to endothelial disfunction. In addition, an increased production of proinflammatory factors along with the induction of apoptosis or necrosis occur, and thereby promote tissue damage and the development of diabetic complications, such as nephropathy, retinopathy, or neuropathy [91]. The development of pancreatic cancer in diabetes is also influenced by obesity, as this state is responsible for the formation of chronic inflammation. Obesity is a chronic inflammatory condition, and its source is visceral adipose tissue, where immune cells, through the secretion of proinflammatory cytokines can induce insulin resistance, and this can further lead to the development of pancreatic cancer [92]. Thus, tumorigenesis in diabetic patients involves adipocyte-derived chronic inflammation. Adipose tissue is hyperplazied and hypertrophied in obese individuals and secrete large amounts of the proinflammatory interleukin Il-1B, thereby recruiting neutrophils associated with tumor (TANs), which induce the activation of pancreatic stellate cells, enhance desmoplasia, and promote pancreatic cancer progression [93]. Obesity has been shown to be associated with increased levels of PLGF (placenta growth factor) and the PLGF/VEGFR-1 system, which is responsible for the modulation of angiogenesis and the promotion of the recruitment of TAMs (tumor-associated macrophages) [92]. TAMs are responsible for secreting tumor-inducing factors, such as EGF (epidermal growth factor), thereby creating an immunosuppressive environment, and promoting angiogenesis, thereby maintaining tumor-related inflammation and inducing metastasis [94]. 

Common risk factors for diabetes and pancreatic cancer also include genetic mutations. GWAS (genome-wide association studies) have revealed that mutations in genes connected with embryonic development and the regulation of pancreas-specific genes such as *Nr5a3*, *HNF1A* (hepatocyte nuclear factor 1), and *PDX1* (pancreatic and duodenal homeobox 1) are associated with the development of pancreatic cancer. Moreover, mutations in the *HFN1A* and *HNF1B* genes associated with type 2 diabetes have also been found to be engaged in pancreatic cancer development [85]. These genes are essential not only for normal pancreatic development, but also for proper β-cell functioning and insulin secretion [82]. Further identification of mutations in the genes that link diabetes and pancreatic cancer is needed. 

On the other hand, diabetes can be a consequence of pancreatic diseases, including cancer. The primary disorders induces damage to the pancreatic islets and further cause the development of diabetes. Compared to type 1 and type 2 diabetes, in type 3c diabetes, not only β-cells but also α-cells are damaged, resulting in the impaired secretion of incretins, insulin resistance, and impaired gluconeogenesis [95]. The most common cause of type 3c diabetes is both acute and chronic pancreatitis, as well as the surgical resection of this organ, hemochromatosis, congenital pancreatic agnesia, and, of course, pancreatic cancer [96]. The most likely pathomechanism of type 3c diabetes is progressive inflammation, which leads to fibrosis and the subsequent hardening of the tissue, and thus reduces not only the number of pancreatic islets, but also limits their functionality. Both the treatment and diagnosis of secondary diabetes is difficult, and involves not only glycemic control by insulin use, but also the treatment of exocrine pancreatic insufficiency through fatty acid enzyme substitutions [97]. 

microRNAs are also altered in both type 1 and type 2 diabetes, and their downregulation or upregulation is often associated with insulin resistance or insulin deficiency. An example of such a molecule is miR-144, whose overexpression disrupts insulin signaling by the inhibition of the expression of the insulin 1 receptor substrate mRNA [98]. Another example is miR-148, which downregulates the glucose transporter *GLUT 1* mRNA [99]. Interestingly, it was found that miR-148, miR-144, and several other microRNAs, including miR-155, miR-24, and miR-122 are associated with the processes leading to the initiation of cancer transformation. miR-155, when altered in diabetes, affects the insulin sensitivity in the liver, adipose tissue, and skeletal muscle. It is also responsible for disorders in the DNA repair process through the inhibition of proapoptotic *TP53* mRNA, and thereby negatively regulates apoptosis [64,100]. It was also proven that miR-144 also contributes to aggravatedly high glucose-induced ROS formation and oxidative stress, which promotes tissue damage and is one of the risk factors for the development of cancer [101]. In turn, miR-24 is also dysregulated in diabetes, and its reduced level is conductive to one of the processes of carcinogenesis, namely EMT. Downregulation of this microRNA increases E-cadherin and β-catenin mRNA, which are involved in EMT, and allows cells to detach from cell agglomerations and to migrate [102]. 

Changes in microRNA expression also contribute to inflammation. A case-control study conducted by Zeinali et al. showed that diabetic patients had elevated levels of miR-122 compared to healthy patients, and this positively correlated with high levels of proinflammatory cytokines, including IL-6 and TNF-α [103]. In turn, inflammation and the accompanying release of proinflammatory cytokines may cause the activation of other microRNAs, such as miR-146, which leads to the downregulation of the transcriptional factor SOX7, and thereby increases the proliferation of cancer cells and supports the process of carcinogenesis as a result [104]. 

microRNAs dysregulated in diabetes are also involved in progression and metastasis. The study conducted by Dai et al. identified in serum six microRNAs common to patients with newly diagnosed diabetes mellitus and pancreatic cancer (miR-483-5p, miR-19a, miR-29a, miR-20a, miR-24, and miR-25, respectively) [105]. miR-19a increases tumor progression by targeting *PTEN* participating in the PI3K/AKT signaling pathway. It was found to be associated with lymph node metastases in patients with pancreatic cancer [106]. 

In summary, more research is needed do show and understand the association between the altered microRNAs in diabetes, which may lead to the development of pancreatic cancer. The discovery of dysregulated microRNAs in both diseases, or separately in diabetes or pancreatic cancer may serve as a marker for the early prediction of the disease. However, it should be clearly underlined that altered microRNAs in diabetes affect many processes connected with cancer development, such as insulin resistance/insulin signaling dysregulation, inflammation, and oxidative stress. Thus, they are able to initiate a cascade of subsequent mechanisms leading to disturbances in the regulation of the cell cycle and apoptosis, meaning they may contribute to the process of cancer development. So far several microRNAs, including miR-483-5p, miR-19a, miR-29a, miR-20a, miR-24, and miR-25 have been demonstrated to be changed in both diabetes and cancer, thereby presenting the link between these two diseases. However, microRNAs altered only in diabetes may also trigger processes leading to tumorigenesis (via other microRNAs), meaning there is possibility that not all microRNAs altered in diabetes will also directly occur in cancer patients, including in pancreatic cancer.

### 3.2. Changes in microRNA Expression Common to Diabetes and Pancreatic Cancer

Pancreatic cancer is known to be one of the worst prognosing cancers. In turn, diabetes is not only a plague of the current world but is also one of the pancreatic cancer risk factors. Among numerus shared mechanisms responsible for the coexistence of these two diseases, alterations in microRNA expression have been intensively explored. microRNAs, through targeting multiple genes, are involved in cellular processes connected with cell cycle control, proliferation, and apoptosis. These processes involves multiple signaling pathways, including the MAPK/KRAS, JAK/STAT, and Wnt/β-catenin pathways [107]. Some of microRNAs present in people with diabetes and pancreatic cancer exhibited an abnormal expression in plasma and tissues [108,109]. miR-21 was found to be involved in the pathogenesis of both diabetes and pancreatic cancer. Studies have shown that miR-21 is upregulated in the tissues of pancreatic cancer patients. It has been found to promote EGF-induced cell cycle progression, proliferation, and inhibition of apoptosis of pancreatic cancer via targeting of the MAPK/ERK and Pi3K/AKT signaling pathways [110]. In patients with diabetes, the levels of this microRNA was also elevated and was demonstrated as a biomarker for diabetic nephropathy [111,112]. It has been shown that patients with type 2 diabetes have elevated plasma levels of miR-221, and in patients with pancreatic cancer, the overexpression of this microRNA was associated with a poorer prognosis and was subsequently recommended as a marker not only for the detection of pancreatic cancer, but also for monitoring tumor dynamics [113,114]. microRNA profiling in patients with prediabetes and type 2 diabetes showed a reduced expression of miR-23 [115]. In turn, the overexpression of the same microRNA was associated with the invasion of pancreatic ductal adenocarcinoma cells by the downregulation of *APAF-1* (apoptotic protease activating factor 1) [116]. A systematic review performed by Dehghanian et al. identified 149 microRNAs that may be involved in the development of type 2 diabetes as well as pancreatic cancer [117]. miR-29 levels are altered in both diabetic and pancreatic cancer patients. Its overexpression occurs in patients with obesity, inhibits insulin-stimulated glucose uptake in adipocytes, and negatively regulates gluconeogenesis in hepatocytes. Studies conducted on young men with diabetes have shown that the expression of miR-29 after exercise increased significantly in skeletal muscle, which is associated with a dysregulated glucose metabolism and insulin resistance [118]. miR-29 in pancreatic cancer reduces expression of miR-20, thereby causing the activation of genes such as *LOXL2*, *CLDN1*, and *MYBL2*, which promote carcinogenic mechanisms [119]. Increased expression of miR-217 has been shown to occur in patients with type 2 diabetes accompanied with diabetic kidney disease. It contributes to proteinuria and renal fibrosis [120]. In contrast, the decreased expression of this microRNA resulted in an increased expression of the *ATAD2* gene, which is involved in pancreatic cancer invasion and drug resistance [121]. Researchers have shown that miR-483-3p expression was elevated in the aortic walls of patients with type 2 diabetes, as well as in endothelial-supportive M2-type macrophages [122]. This contributes to increased endothelial apoptosis and *in vitro* endothelialization disorders, which consequently lead to an impaired trauma-induced vascular response in people with diabetes [122]. Overexpression of miR-483-3p occurs not only in pancreatic cancer, but also in precancerous lesions. This non-coding RNA promotes the migration and invasion of pancreatic cancer cells. An overview of some of the most common microRNAs with altered expression is presented in Table 3. 

Interestingly, metformin, a drug of the first choice in type 2 diabetes treatment, has been demonstrated to affect the expression of several microRNAs. Metformin inhibits glucose production in the liver and increases the uptake of glucose by peripheral tissues, including the skeletal muscle [123]. The use of metformin by patients with type 2 diabetes has been shown to reduce the risk of cancer overall [124]. In addition, studies on hamsters have shown that metformin reduces the incidence of pancreatic cancer [124]. It was detected that metformin blocks the EMT and inactivates CSCs (cancer stem cells), thereby reducing chemoresistance [125,126]. Further studies have shown that these changes were caused by the restoration of the expression of the following microRNAs: let-7a, let-7b, let-7c, miR-26a, miR-101, miR-200b, and miR-200c, respectively [127]. Increased expression of the let-7 family microRNAs inhibits the expression of the *Rac* and *c-myc* oncogenes in malignant cells and reduces tumor aggressiveness [128]. In turn, the increased expression of miR-26a and miR-101 resulted in the repression of the EXH2 protein responsible for angiogenesis [127].

Taken together, microRNAs participate in multiple cellular processes. Therefore they can be used not only as markers for the early prediction or prognosis of pancreatic cancer, but they can also be used as therapeutic targets for the evaluation of the chemoresistance of cancer cells. 

### 3.3. The Real Clinical Application of the Discoveries on microRNAs in Pancreatic Cancer 

microRNAs are small, non-coding molecules that are key regulators of gene expression. It has been shown that their altered levels, both upregulated and downregulated, correlate with many diseases, including pancreatic cancer. It has been suggested that they may be useful biomarkers for the early detection of the disease, as tools to monitor the course of the disease, and even be utilized for therapy. However, their real clinical application is limited. Unfortunately, there are still too few clinical studies that confirm the effectiveness and usefulness of microRNAs or antagomirs in pancreatic cancer treatment. The majority of data present microRNA action in pancreatic cancer derived from in vitro studies and animal models of pancreatic tumor, while there is only a few studies involving people. There is also a problem with the delivery of microRNA molecules to the target place in the body. Although microRNAs are small in size, they are nucleic acids, and their application is associated with an array of hurdles with respect to in vivo delivery [129]. It is therefore necessary to create a delivery system that will allow microRNA molecules to overcome systemic barriers while simultaneously remaining biologically active, stable, and non-toxic to the patient. Therefore, the use of microRNA therapy at the current stage may be risky for the patient. However, their utility for the diagnosis and monitoring of the course of pancreatic cancer therapy raises more hope, as more tests are available and there is also a lower risk for the patient.

## 4. Conclusions

Recently, scientists have been devoting a lot of attention to non-coding RNAs, including lncRNAs (long non-coding), snRNAs (small nuclear), piRNAs (piwi-interacting) and microRNAs. The latter ones, arising from an originally long RNA sequence, are processed into a short but functional transcript of about 22 nucleotides long, and are able to target hundreds of genes, often acting specifically for a given cell or tissue [130]. As a result, great hopes are being placed on this non-coding RNA due to its ability to regulate not only single elements, but entire biological pathways, bringing with it immense therapeutical potential. 

In both pancreatic cancer and diabetes, changes in microRNA expression were observed, including miR-21, which can be a predictor of chemoresistance, and at the same time indicate diabetic nephropathy. miR-155 and miR-23, oncogenic regulators of pancreatic cancer, are perfect as predictors of prediabetes.Their altered level is often associated with a poor patient prognosis, chemoresistance, and increased invasion and metastasis. Analysis of the expression changes of these specific microRNAs allows for the early detection of pancreatic cancer, and thereby allows for the earlier implementation of therapy. However, microRNAs cannot serve only as biomarkers, but they can also bring significant effects in treatment, such as miR-26a or miR-101, whose expression restoration allows to limit the invasion of cancer cells and cause the partial abolition of resistance to chemotherapeutic agents, thereby significantly improving the prognosis for patients. However, further research is needed to identify other microRNAs not only associated with pancreatic cancer, but also with diabetes, which is one of the most important risk factors for this cancer. 

## Figures and Tables

**Figure 1 ijms-24-10252-f001:**
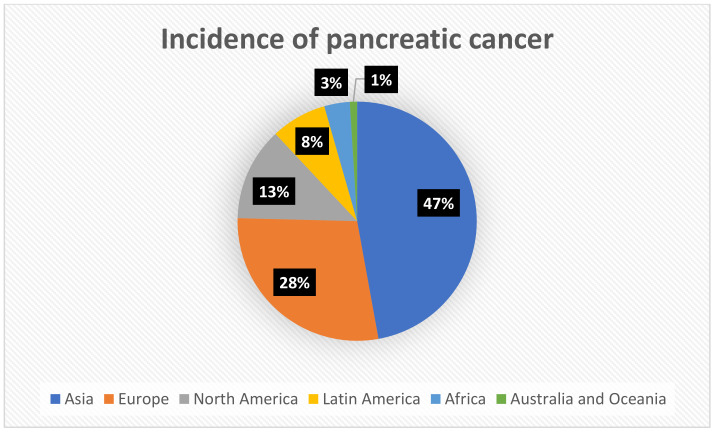
The percentage of patients suffering from pancreatic cancer across the continents. Data obtained from GLOBOCAN 2020.

**Figure 2 ijms-24-10252-f002:**
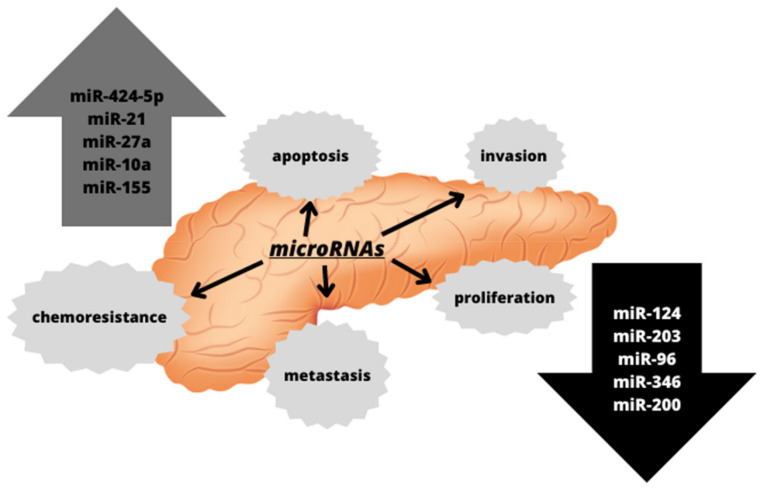
Altered microRNAs detected in pancreatic cancer and their impact on cellular processes associated with carcinogenesis.

**Figure 3 ijms-24-10252-f003:**
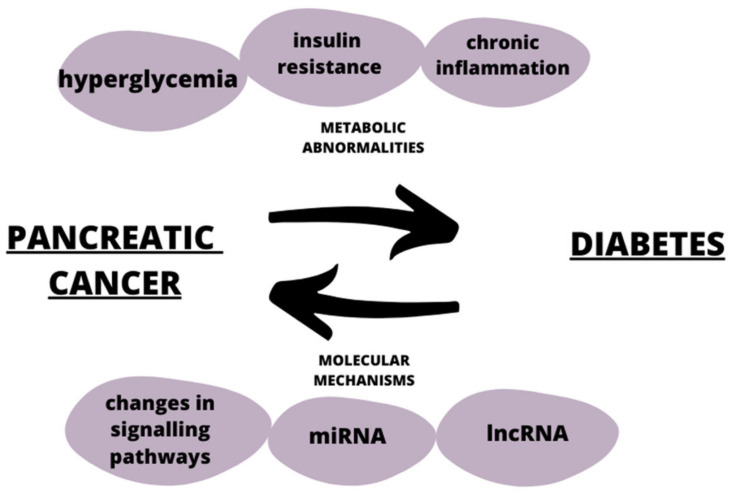
The relationship between pancreatic cancer and diabetes.

**Table 1 ijms-24-10252-t001:** Modifiable and non-modifiable risk factors of pancreatic cancer.

Modifiable Risk Factors	Non-Modifiable Risk Factors
Smoking	Age
Alcohol drinking	Family history of the disease
Improper diet (obesity)	Genetic mutations
Prediabetes/insulin resistance	Diabetes mellitus

**Table 2 ijms-24-10252-t002:** Types of diabetes and their characteristics, based on the World Health Organization Classification of Diabetes Mellitus, 2019.

Types of Diabetes	Characteristics
Type 1 insulin-dependent diabetes	β-cell destruction and absolute insulin deficiency, onset most common in childhood, and it can be idiopathic as well as genetically determined.
Type 2 non-insulin-dependent diabetes	Insulin resistance, commonly associated with being overweight and obesity, and is most common in adults.
Type 3 secondary diabetes	It is a consequence of, among others, chronic or acute pancreatitis, pancreatectomy, pancreatic cancer, or other diseases, and is often confused with type II diabetes.
Diabetes mellitus in pregnancy	Usually occurs in the second and third trimesters of pregnancy, may persist beyond the end of pregnancy, and is associated with a significant risk to both the mother and newborn.
Different types of diabetes: neonatal diabetes, MODY-type diabetes, diseases of the exocrine pancreas, drug-induced, and infection-related	Drug-induced: some medicines, such as thiazides or pentamidines can destroy β-cells or lead to insulin resistance. MODY diabetes: occurs in young people, leads to impaired insulin secretion, and is inherited in an autosomal dominant manner. Neonatal diabetes: monogenic form, and can be both temporary and permanent. Infection-related: cytomegalovirus or Coxsackie B virus is able to induce type I diabetes.

**Table 3 ijms-24-10252-t003:** microRNAs associated with diabetes and pancreatic cancer (***↑*** upexpression, ***↓*** downexpression).

microRNA	Pancreatic Cancer		Diabetes		References
	Change of Expression (↑/↓)	Function	Change of Expression (↑/↓)	Function	
miR-145	*↓*	Targeting of key proteins of insulin signaling, such as IRS-1 and AKT.	*↓*	Suppression of oncogene expression, including ANG-2 and NEEDD9.	[117]
miR-155	*↓*	Component of the response of macrophages and monocytes to LPS and TNF-α.	*↑*	Promoting the development and invasion of pancreatic cancer by targeting p53.	[117]
miR-21	*↓*	Development and proper functioning of the endocrine pancreas, regulation of the insulin secretion, angiogenesis, and modulation of the inflammatory response.	*↑*	Negative regulation of the tumor suppressor gene PTEN.	[117]
miR-181a	*↑*	Lowering the level of the SIRT1 protein, which is responsible for carbohydrate and lipid metabolism.	*↑*	Negative regulation of the PI3K-AKT pathway, leading to excessive cell proliferation.	[117]
miR-148a	*↑*	Targeting of the cholecystokinin receptor 2 gene, leading to the promotion of diabetic obesity by increasing the neuropeptide Y.	*↓*	Inhibition of ASPC-1 cell proliferation and metastasis.	[117]
miR-571	↑	Contributes to renal fibrosis and diabetic nephropathy.	↑	Targets guanylate 2-binding protein.	[117]
miR-29	*↓*	Metastasis, tumorigenesis, and progression of cancer.	↑	Regulator of skeletal muscle metabolism.	[118,119]
miR-217	*↓*	Regulating pancreatic cancer progression by ATAD2 gene targeting.	↑	Development of renal fibrosis and proteinuria.	[120,121]
miR-483-3p	↑	Promoting migration and invasion of the cancer cells.	↑	Increased endothelial apoptosis.	[122]
miR-144	*↓*	Migration, proliferation, and invasion of pancreatic cells.	*↑*	Disrupting insulin signaling by inhibiting the insulin 1 receptor substrate.	[98]
miR-24	*↓*	Promoting the malignancy and EMT process.	↑	Inhibition of β-cell proliferation and insulin secretion.	[102]

## Data Availability

Data sharing not applicable.

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
