# Peer review of "The Link between Diabetes, Pancreatic Tumors, and miRNAs—New Players for Diagnosis and Therapy?"

_ijms, 2023, doi:10.3390/ijms241210252_

Round 1

Reviewer 1 Report

The work is well written and the association between diabetes and pancreatic cancer is interesting. I have two minor observations before its publication: 1) Please revise the order of the cell signaling pathways and 2) Please exchange references 3, 18, 30, 41, 45, 46, 51, 55, 60 to english because they are in Polish language. 

Author Response

Review Report 1

Thank you for your work and time to read this manuscript and point out valuable comments. The manuscript has been enriched with information according to the instructions of the reviewers. Were made in review mode. We hope that the prepared changes are sufficient and in its current form, the manuscript meets the expectations of the reviewers because in the opinion of the authors, its quality has improved significantly.

Point 1: Please revise the order of the cell signaling pathways

Answer: In chapter 2, the signaling pathways were changed according to the sequence of succession during carcinogenesis (initiation, progression, metastasis). (lines 257-292)

Point 2: Please exchange references 3, 18, 30, 41, 45, 46, 51, 55, 60 to English because they are in Polish language

Answer: All indicated  references has been changed to relevant original paper published in English.

Reviewer 2 Report

This review paper aims to provide an overview of the link between diabetes, pancreatic tumor, and microRNAs (miRNAs). The abstract states the roles of several miRNAs in both of these two diseases. However, the knowledge of miRNAs discussed in this paper, which establishes a link between diabetes and pancreatic tumors, is based on a limited number of papers. Consequently, this paper does not offer a comprehensive review from the perspective of miRNAs.

1.     Line 54, “microRNa” ” microRNA”

2.     microRNA (miRNAs) was stated in Line 54. After that, “miRNAs” should be used instead of “microRNAs”. But “microRNAs” was used throughout the paper.

3.     Line 70  “slight advantage among men” is not clear.

4.     Line 72. Are there any reasons why most cases occurred in Asia and Europe?

5.     This paper reviews the backgrounds of diabetes and pancreatic tumor including the epidemiology, pathophysiology, and diagnosis and therapy of these diseases.

However, the discussion on miRNAs linking both diseases is very limited. Many miRNAs are involved in the pathophysiology of diabetes, but this study lacks a comprehensive review.

6.     Table 3 listed the miRNAs linking diabetes and pancreatic tumor. However, this result is only based on one paper (Reference 91). This does not provide a comprehensive review of miRNAs related to diabetes and pancreatic tumor.

This review paper aims to provide an overview of the link between diabetes, pancreatic tumor, and microRNAs (miRNAs). The abstract states the roles of several miRNAs in both of these two diseases. However, the knowledge of miRNAs discussed in this paper, which establishes a link between diabetes and pancreatic tumors, is based on a limited number of papers. Consequently, this paper does not offer a comprehensive review from the perspective of miRNAs.

1.     Line 54, “microRNa” ” microRNA”

2.     microRNA (miRNAs) was stated in Line 54. After that, “miRNAs” should be used instead of “microRNAs”. But “microRNAs” was used throughout the paper.

3.     Line 70  “slight advantage among men” is not clear.

4.     Line 72. Are there any reasons why most cases occurred in Asia and Europe?

5.     This paper reviews the backgrounds of diabetes and pancreatic tumor including the epidemiology, pathophysiology, and diagnosis and therapy of these diseases. However, the discussion on miRNAs linking both diseases is very limited. Many miRNAs are involved in the pathophysiology of diabetes, but this study lacks a comprehensive review.

6.     Table 3 listed the miRNAs linking diabetes and pancreatic tumor. However, this result is only based on one paper (Reference 91). This does not provide a comprehensive review of miRNAs related to diabetes and pancreatic tumor.

Reviewer 3 Report

The authors presented an interesting review summarizing the contingent contribution of miRNAs in both pancreatic cancer and diabetes. The authors have added some interesting figures/patterns to recap the take-home message to scientists. The review appears to be well organized and easy to read. However, we would like to suggest to the authors some additional things that could increase the solidity of the manuscript.

So far, we'd like to capture the attention of authors by inviting them to look for these important hot topics:

Major revision:

1) The authors could add an important paragraph, in which to discuss the important role of DICER expression in pancreatic cancer and its effects on miRNA modulation.

2) Both miRNAs 101 and 155 are closely involved in IPMN pathologies. Can the authors add any important insights into the "triage" that includes PDACs, IPMNs, and miRNAs?

3) Looking both at the title of the manuscript and within the main body of the manuscript, two chapters seem to be missing: the first concerning the relationship between the role of miRNAs and therapeutic approaches; the second concerns the real clinical application of the discoveries on mIRNAs in PDAC. We would like to invite the authors to trace these two parts to connect the manuscript content to the conclusions.

Minor comment:

1) Reference 36 editing style, is not well formatted.

No additional comments we have.

Round 2

Reviewer 2 Report

The authors have addressed most of my comments. I only have a minor comment. Table 3 should include the references for miRNAs. 
